# HumBugDB: A Large-scale Acoustic Mosquito Dataset

**Ivan Kiskin**[*]
University of Oxford

**Marianne Sinka**[†]
University of Oxford

**Adam D. Cobb**[‖]
SRI International

**Waqas Rafique**[*]
University of Oxford

**Lawrence Wang**[*]
University of Oxford

**Davide Zilli**[¶]
Mind Foundry Ltd

**Benjamin Gutteridge**[*]
University of Oxford

**Rinita Dam**[†]
University of Oxford

**Theodoros Marinos**[††]
University of Surrey

**Yunpeng Li**[††]
University of Surrey

**Dickson Msaky**[‡]
IHI Tanzania

**Emmanuel Kaindoa**[‡]
IHI Tanzania

**Gerard Killeen**[§]
UCC, BEES

**Eva Herreros-Moya**[†]
University of Oxford

**Kathy Willis**[†]
University of Oxford

**Stephen J. Roberts**[*]
University of Oxford

[*]{ikiskin, waqas, beng, sjrob}@robots.ox.ac.uk, lawrence.wang@eng.ox.ac.uk
[†]{marianne.sinka, kathy.willis, rinita.dam, eva.herreros-moya}@zoo.ox.ac.uk,
[‖]adam.cobb@sri.com, [††]{tm00591, yunpeng.li}@surrey.ac.uk, [§]gerard.killeen@ucc.ie,
[¶]davide.zilli@mindfoundry.ai, [‡]{dmsaky,ekaindoa}@ihi.or.tz.

## Abstract

This paper presents the first large-scale multi-species dataset of acoustic recordings of mosquitoes tracked continuously in free flight. We present 20 hours of audio recordings that we have expertly labelled and tagged precisely in time. Significantly, 18 hours of recordings contain annotations from 36 different species. Mosquitoes are well-known carriers of diseases such as malaria, dengue and yellow fever. Collecting this dataset is motivated by the need to assist applications which utilise mosquito acoustics to conduct surveys to help predict outbreaks and inform intervention policy. The task of detecting mosquitoes from the sound of their wingbeats is challenging due to the difficulty in collecting recordings from realistic scenarios. To address this, as part of the HumBug project, we conducted global experiments to record mosquitoes ranging from those bred in culture cages to mosquitoes captured in the wild. Consequently, the audio recordings vary in signal-to-noise ratio and contain a broad range of indoor and outdoor background environments from Tanzania, Thailand, Kenya, the USA and the UK. In this paper we describe in detail how we collected, labelled and curated the data. The data is provided from a PostgreSQL database, which contains important metadata such as the capture method, age, feeding status and gender of the mosquitoes. Additionally, we provide code to extract features and train Bayesian convolutional neural networks for two key tasks: the identification of mosquitoes from their corresponding background environments, and the classification of detected mosquitoes into species. Our extensive dataset is both challenging to machine learning researchers focusing on acoustic identification, and critical to entomologists, geo-spatial modellers and other domain experts to understand mosquito behaviour, model their distribution, and manage the threat they pose to humans.

# 1 Introduction

There are over 100 genera of mosquito in the world containing over 3,500 species and they are found on every continent except Antarctica [Harbach, 2013]. Only one genus (*Anopheles*) contains species capable of transmitting the parasites responsible for human malaria. *Anopheles* contain over 475 formally recognised species, of which approximately 75 are vectors of human malaria, and around 40 are considered truly dangerous [Sinka et al., 2012]. These 40 species are inadvertently responsible for more human deaths than any other creature. In 2019, for example, malaria caused around 229 million cases of disease across more than 100 countries resulting in an estimated 409,000 deaths [World Health Organization, 2020]. It is imperative therefore to accurately locate and identify the few dangerous mosquito species amongst the many benign ones to achieve efficient mosquito control. Mosquito surveys are used to establish vector species' composition and abundance, human biting rates and thus the potential to transmit a pathogen. Traditional survey methods, such as human landing catches, which collect mosquitoes as they land on the exposed skin of a collector, can be time consuming, expensive, and are limited in the number of sites they can survey. They can also be subject to collector bias, either due to variability in the skill or experience of the collector, or in their inherent attractiveness to local mosquito fauna. These surveys can also expose collectors to disease. Moreover, once the mosquitoes are collected, the specimens still need to undergo post sampling processing for accurate species identification. Consequently, an affordable automated survey method that detects, identifies and counts mosquitoes could generate unprecedented levels of high-quality occurrence and abundance data over spatial and temporal scales currently difficult to achieve. We therefore utilise low-cost smartphones, acting as acoustic mosquito sensors, to solve this task. The exponential increase in smartphone ownership is a worldwide phenomenon. Governments and independent companies are continuing to extend connectivity across the African continent [Friederici et al., 2017]. More than half of sub-Saharan Africa is expected to be connected to a mobile service by 2025 [GSMA, 2020]. With this expanding coverage of mobile phone networks across Africa, there is an emerging opportunity to collect huge datasets, as exemplified by the World Bank's Listening to Africa Initiative [World Bank Organisation, 2017]. Our target application (Section 3) uses a free downloadable app, which means that every smartphone can be a mosquito monitor.

**Our contribution** In order to assist research in methods utilising the acoustic properties of mosquitoes, as part of the HumBug project (described in Section 3) we contribute:

- **Data:** http://doi.org/10.5281/zenodo.4904800: A vast database of 20 hours of finely labelled mosquito sounds, and 15 hours of associated non-mosquito control data, constructed from carefully defined recording paradigms. Data was collected over the course of five years in a global collaboration with mosquito entomologists. Recordings were captured from 36 species with a mix of low-cost smartphones and professional-grade recording devices, to capture both the most accurate noise-free representation, as well as the sound that is likely to be recorded in areas most in need. A diverse quantity of wild and lab culture mosquitoes is included in the database to capture the biodiversity of naturally occurring species. Our data is stored and maintained in a PostgreSQL database, ensuring label correctness, data integrity, and allowing efficient updates and re-release of data.

- **Mosquito event detector and species classification baselines:** https://github.com/HumBug-Mosquito/HumBugDB: Detailed tutorial code for training state-of-the-art Bayesian neural network models for two key tasks – Mosquito Event Detection (MED): distinguishing mosquitoes of any species from their background surroundings, such as other insects, speech, urban, and rural noise; Mosquito Species Classification (MSC): species classification of over 1,000 individually captured wild mosquitoes. In combination, our tasks and models are the first of their kind to use large-scale real-world data for the purpose of automating acoustic mosquito species monitoring.

The rest of the paper is structured as follows. Section 2 details related datasets and describes how ours contributes to the literature uniquely. Section 3 shows the intended use cases for the data and models released in this paper. Section 4 describes in depth the sources and collection methods of data present. The steps taken to benchmark models for MED and MSC are given in Section 5. We discuss the results that our models achieve, and the open challenges remaining. We conclude in Section 6.

Comprehensive instructions for using our baseline models and feature extraction code are provided in Appendix B, and additional details on all the metadata in Appendix C. The datasheet (Appendix

D) details the dataset's composition (D.2), acquisition process (D.3), preprocessing (D.4), past and suggested use cases (D.5), data bias and mitigation strategies (D.6), and maintenance policies (D.7).

## 2   Related work

Mosquitoes have particularly short, truncated wings allowing them to flap their wings faster than any other insect of equivalent size – up to 1,000 beats per second [Simões et al., 2016, Bomphrey et al., 2017]. This produces their distinctive flight tone and has led many researchers to try and use their sound to attract, trap or kill them [Perevozkin and Bondarchuk, 2015, Johnson and Ritchie, 2016, Jakhete et al., 2017, Joshi and Miller, 2021]. Table 1 provides details of the few datasets released to the public to aid this research. We discuss the varying sensor modalities separately, due to their inherent differences in properties.

Table 1: Publicly available datasets. *'Average mosquito'* is the approximate length of audible mosquito recording per sample. Where not known, *'Mosquito'* is estimated from the average mosquito sample duration multiplied by the number of positive samples. *'Type'* represents wild captured or lab grown mosquitoes (in order of prevalence). Crowdsourced recordings or labels are marked with (*).

| Dataset | Sensor | Mosquito (Background) | Average mosquito | Species | Type |
|---------|--------|----------------------|------------------|---------|------|
| Chen et al. [2014, UCR] | Opto-acoustic | 17 min (N/A) | $\approx 0.02$ s | 6 | Lab |
| Fanioudakis et al. [2018] | Opto-acoustic | 39 hr (N/A) | $\approx 0.5$ s | 6 | Lab |
| Vasconcelos et al. [2020] | Acoustic | 15 min (N/A) | 0.3 s | 3 | Lab |
| Mukundarajan et al. [2017] (*) | Acoustic | N/A (N/A) | N/A | 20 | Lab + wild |
| Kiskin et al. [2019, 2020] (*) | Acoustic | 2 hr (20 hr) | 1 s | N/A | Lab + wild |
| **HumBugDB** | Acoustic | 20 hr (15 hr) | 9.7 s | 36 | Wild + lab |

**Opto-acoustic approaches**   *'Wingbeats'* [Fanioudakis et al., 2018] and *'UCR Flying Insect Classification'* [Chen et al., 2014] are datasets collected via optical sensors with high signal-to-noise-ratio (SNR). We note this is a different, but complementary, approach. Due to the directionality of the recording method, typical sample durations are encountered from "only a few hundredths of a second" [Chen et al., 2014] to approximately half a second [Fanioudakis et al., 2018]. The approach therefore does not capture the acoustical properties of mosquito sound in free flight which aid mosquito detection in purely acoustic approaches [Vasconcelos et al., 2020]. Furthermore, these datasets survey lab-grown mosquito colonies which do not capture the biodiversity of mosquitoes encountered in the wild [Huho et al., 2007, Hoffmann and Ross, 2018].

**Acoustic approaches**   Vasconcelos et al. [2020] motivated their release by stating that none of the published datasets include environmental noise, which is essential to fully characterise mosquitoes in real-world scenarios. The dataset consists of 300 ms snippets, amounting to 15 minutes of recordings. This is an excellent first step. However, for deep learning algorithms the dataset is not readily useable due to its size. Moreover, state-of-the-art models for acoustic classification use training example sizes of at least 0.96 seconds for a variety of audio event detection tasks [Hershey et al., 2017, Pons et al., 2017, Shimada et al., 2020]. Our dataset consists of mosquito samples with an average duration of 10 seconds. Additionally, we supply equal quantities of background collected in the same controlled conditions to form a balanced class distribution of mosquito occurrences and a negative control group (see Section 4). This is to prevent the recording device or background environment from becoming a confounding factor for the detection of acoustic events [Coppock et al., 2021].

Mukundarajan et al. [2017] released an acoustic dataset recorded in free flight with smartphones. However, due to a lack of a rigorous protocol, the quality of the recordings is inconsistent, and there is a lack of metadata recording external factors which influence mosquito sound. There are no labels to timestamp the mosquito events in files where mosquito sound is only sporadic, detracting from the overall utility of the dataset.

Kiskin et al. [2019, 2020] released 22 hours of audio, with crowdsourced labels covering overlapping two-second sections. However, of these, only 2 hours were labelled as containing mosquito sound. In addition, the accuracy of the labels was unknown, and the task of labelling was made difficult as clips

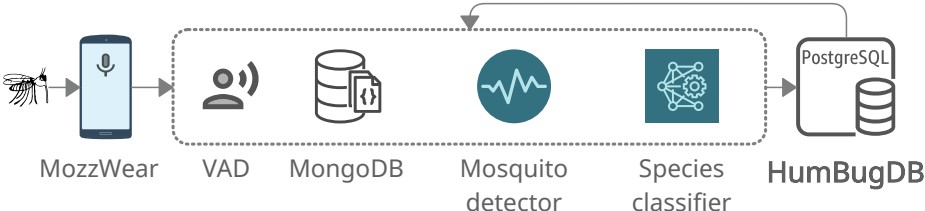

Figure 1: Target workflow. Our mobile phone app, MozzWear, captures audio. The app synchronises to a central server (dashed). Voice activity is removed and data is stored in a MongoDB instance. Audio undergoes mosquito event detection (MED) and subsequent species classification (MSC). Successful detections are used to update HumBugDB. Information feeds back to improve the model.

were presented in isolation, lacking the relevant background information that specialists utilised for their labels. Curated data of that release is a subset of HumBugDB, in which we improve upon the past release thanks to a joint effort between the zoological and machine learning communities.

Nevertheless, we stress that experimentation which combines information from all of the datasets found in the literature is highly encouraged, and may help find solutions that cover multiple recording modalities, such as both opto-acoustic and acoustic sensors.

## 3 Data for mosquito-borne disease prevention

The HumBug project is a collaboration between the University of Oxford and mosquito entomologists worldwide [HumBug, 2021]. One of the goals of the project is to develop a mosquito acoustic sensor that can be deployed into the homes of people in malaria-endemic areas to help monitor and identify the mosquito species, allowing targeted and effective vector control. In the following paragraphs we describe the system of Figure 1 to be deployed for this purpose, the role of each component, and the two key tasks (MED, MSC) our models are able to address thanks to the data of HumBugDB.

**Capturing mosquito with smartphones**   We developed a power-efficient app to record mosquito flight tone using the in-built microphone on a smartphone (MozzWear [Marinos et al., 2021]). We used 16-bit mono PCM wave audio sampled at 8,000 Hz, based on prior acoustic low-cost smartphone recording solutions for mosquitoes [Li et al., 2017b, Kiskin et al., 2018]. To ensure mosquitoes fly close enough to a smartphone, we have developed an adapted bednet (the *HumBug Net*) that exploits the inherent behaviour of host-seeking mosquitoes (Figure 2, for details refer to Sinka et al. [2021, Sec. 2.1.2]). The combination of the bednets and smartphones constitutes the intended use case, for which we construct MED: Test A (see Table 2).

**MongoDB**   Following app recording, audio is synchronised by the app to a central file server for the storage of sound recordings, and a MongoDB [MongoDB Inc, 2021] instance for the storage of metadata. The server possesses a frontend dashboard where recordings and predictions fed back from the model can be accessed. The unstructured nature of the NoSQL engine allows for additional flexibility in storing metadata, especially when new information becomes available.

**Mosquito Event Detection (MED)**   A Bayesian convolutional neural network (BCNN), which provides predictions with uncertainty metrics [Kiskin et al., 2021] is used to detect mosquito events. Positive predictions are then filtered by the probability, mutual information and predictive entropy [Houlsby et al., 2011], screened, and stored in a curated database. This drastically reduces the time spent labelling by domain experts – for our bednet data recorded in Tanzania, we estimate 1 to 2 % of 2,000 hours of recorded data contained mosquito events. Finding these events without assistance from the model was infeasible due to the vast quantity of data. Section 5.1 defines two test sets to further motivate model development for this task.

**Mosquito Species Classification (MSC)**   A second BCNN is trained specifically for species classification. Once mosquito events have been identified, a probability distribution over species is produced. The report is made available through an HTML dashboard and can be streamed to the app to provide feedback to users. Section 5.2 details the MSC task.

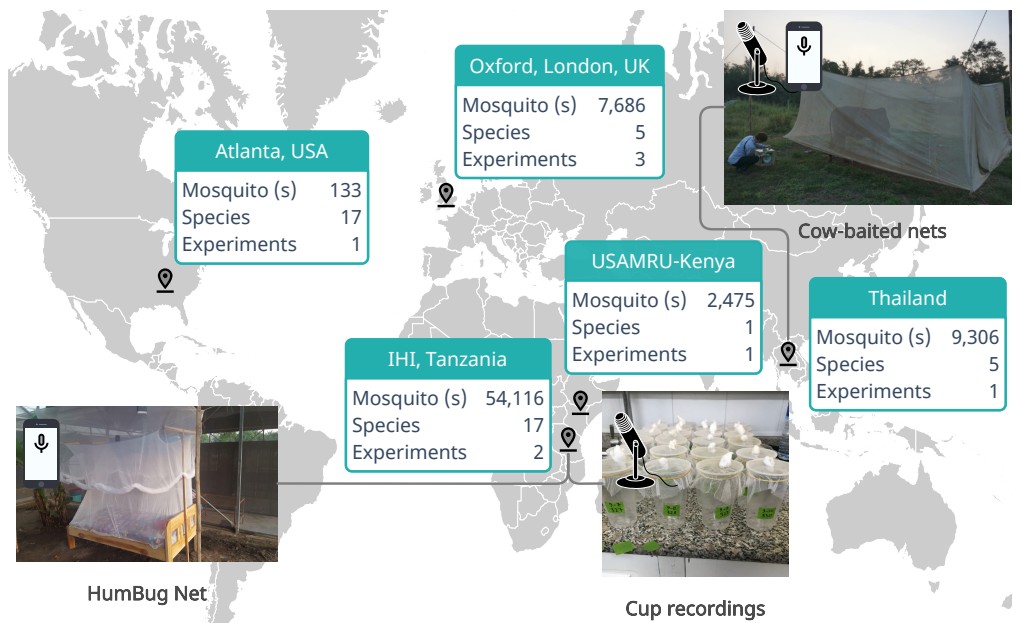

Figure 2: Map of aggregated data acquisition sites. HumBug Net: Sinka et al. [2021, Sec. 2.1.2].

**PostgreSQL database**   Due to the complex requirements of variables and data storage, we designed a relational database [PostgreSQL Global Development Group, 2021] which ensures a standardisation in the labelling and metadata process. This mitigates a major cause of data quality issues and time costs in field studies. Data has been obtained from controlled studies in focused experiments, with the aid of MED models where applicable. We discuss the sources of the data present in Section 4. Recordings are stored in wave format at their respective sample rates, and all the metadata in csv format (Appendix C). For our maintenance policy, details of ethics agreements, and detailed documentation, refer to the datasheet (Appendix D).

**Privacy**   As a subset of data from the database may contain human speech, and other types of personal data, we include in this paper only audio which has been assigned an explicit label of *'mosquito', 'audio', 'background'*, or otherwise full consent from members was obtained (for example where entomology experts state a recording ID). To ensure no speech that has not had explicit consent for is included in future releases, we perform voice activity detection (VAD) and removal using Google's WebRTC project, which is open-source and lightweight [Ali, 2018, Karrer, 2020]. Sahoo [2020] tested the WebRTC VAD method over 396 hours of data, across multiple recording types. The approach was between 77 % and 99.8 % accurate. A list of approved ethical review processes is given in Appendix D.3.

## 4   The HumBugDB dataset

Our large-scale multi-species dataset contains recordings of mosquitoes collected from multiple locations globally, as well as via different collection methods. Figure 2 shows the different locations, with the availability of labelled mosquito sound (in seconds) and number of species, and the number of experiments conducted at each location. In total, we present 71,286 seconds (20 hours) of labelled mosquito data with 53,227 seconds (15 hours) of corresponding background noise to aid with the scientific assessment process, recorded at the sites of 8 experiments. Of these, 64,843 seconds contain species metadata, consisting of 36 species (or species complexes) with the distributions illustrated in Appendix C, Figure 11 and Table 6. Table 2 gives a more detailed summary of the nature of mosquitoes that were captured, and Appendix C gives a complete explanation of every field in the

Table 2: Key audio metadata and division into train/test for the tasks of MED: Mosquito Event Detection, and MSC: Mosquito Species Classification. *'Wild'* mosquitoes captured and placed into paper *'cups'* or attracted by bait surrounded by *'bednets'*. *'Culture'* mosquitoes bred specifically for research. Total length (in seconds) of mosquito recordings per group given, with the availability of species meta-information in parentheses. Total length of corresponding non-mosquito recordings, with matching environments, given as *'Negative'*. Full metadata documented in Appendix C.

| Tasks: Train/Test | Mosquito origin | Site Country | Method (year) | Device (sample rate) | Mosquito (s) (with species) | Negative (s) |
|---|---|---|---|---|---|---|
| MSC: Train/Test MED: Train | Wild | IHI Tanzania | Cup (2020) | Telinga 44.1 kHz | 45,998 45,998 | 5,600 |
| MED: Train | Wild | Kasetsart Thailand | Cup (2018) | Telinga 44.1 kHz | 9,306 2,869 | 7,896 |
| MED: Train | Culture | OxZoology UK | Cup (2017) | Telinga 44.1 kHz | 6,573 6,573 | 1,817 |
| MED: Train | Culture | LSTMH (UK) | Cup (2018) | Telinga 44.1 kHz | 376 376 | 147 |
| MED: Train | Culture | CDC USA | Cage (2016) | Phone 8 kHz | 133 127 | 1,121 |
| MED: Train | Culture | USAMRU Kenya | Cage (2016) | Phone 8 kHz | 2,475 2,475 | 31,930 |
| MED: Test A | Culture | IHI Tanzania | Bednet (2020) | Phone 8 kHz | 4,118 4,118 | 3,979 |
| MED: Test B | Culture | OxZoology UK | Cage (2016) | Phone 8 kHz | 737 737 | 2,307 |
| **Total** | | | | | **71,286 64,843** | **53,227** |

metadata. We also demonstrate example spectrograms for a variety of mosquito species in Figure 8, Appendix B.5, and supply a tool to play back and visualise audio clips[1] (see Figure 9, Appendix B.5).

In the following section we break down the data sources according to the nature of mosquitoes – bred within laboratory culture (Section 4.1.1) or wild (Section 4.1.2). We discuss the recording device and the environment the free-flying mosquitoes were recorded in: culture cages, cups or in HumBug Nets. We also state the methods of capture, where applicable, documented in more detail in Appendix C.

## 4.1 Data collection

### 4.1.1 Laboratory culture mosquitoes

Many institutes that conduct research into mosquito-borne diseases hold laboratory cultures of common vector species. These include primary malaria vectors (e.g. *An. arabiensis*), primary vectors of the dengue virus (*Aedes albopictus*), yellow fever virus (*Aedes aegypti*) and the West Nile virus (*Culex quinquefasciatus*). The controlled conditions of laboratory cultures produce uniformly sized fully-developed adult mosquitoes which are used for a variety of purposes, including trialling new insecticides or examining the genome of these insects.

**UK, Kenya, USA**    Mosquitoes were recorded by placing a recording device into the culture cages where one or multiple mosquitoes were flying, or by placing individual mosquitoes into large cups and holding these close to the recording devices (denoted by device_type). Recordings were captured at the London School of Tropical Medicine and Hygiene (LSTMH), the United States Army Medical Research Unit-Kenya (USAMRU-K), the Center for Diseases Control and Prevention (CDC), Atlanta, as well as with mosquitoes raised from eggs at the Department of Zoology, University of Oxford. We reserve one set of these recordings taken in culture cages by Zoology, Oxford, as MED: Test B (Table 2). Past models were able to achieve excellent mosquito detection performance when trained

---

[1] https://github.com/HumBug-Mosquito/HumBugDB/blob/master/notebooks/spec_audio_multispecies.ipynb

on recordings held out from the same experiment [Kiskin et al., 2018, 2017]. In this paper we treat this experiment as disparate from the remaining data, increasing the difficulty of the detection task.

**Tanzania**    To achieve targeted vector control through the deployment in people's homes, we need to be able to passively capture the mosquito's flight tone. Therefore, in our database we include mosquitoes passively recorded in the Ifakara Health Institute's (IHI) semi-field facility, that most closely resembles the intended use of the HumBug system. It is for this reason that a labelled subset (by an expert zoologist with the help of a BCNN) of this data forms MED: Test A (Table 2). The facility houses six chambers containing purpose-built experimental huts, built using traditional methods and mimicking local housing constructions, with grass roofs, open eaves and brick walls. Four different configurations of the HumBug Net [Sinka et al., 2021], each with a volunteer sleeping under the net, were set up in four chambers. Budget smartphones were placed in each of the four corners of the HumBug Net (Figure 2). Each night of the study, 200 laboratory cultured *An. arabiensis* were released into each of the four huts and the MozzWear app began recording.

### 4.1.2    Wild captured mosquitoes

Wild mosquitoes naturally exhibit far greater variability and are thus crucial to sample for real-world detection capability assessment. To study how this affects our ability to distinguish different species, we conducted experiments in Thailand and Tanzania. Recordings made in Thailand were used to demonstrate that flight tone has the potential to distinguish different species [Li et al., 2018]. In Section 5.2, we consider an extension with a greater number of species and more rigorous experimental design with data recorded in Tanzania, forming the MSC dataset of Table 2.

**Thailand**    Across the malaria endemic world, Asia has more *dominant* vector species (mosquitoes whose abundance or propensity to bite humans makes them particularly efficient vectors of disease) than anywhere else. Mosquitoes were sampled using ABNs (animal-baited nets in Figure 2), human-baited nets (HBNs) and larval collections (LC) over a period of two months during peak mosquito season (May to October 2018). Sampling was conducted in Pu Teuy Village at a vector monitoring station owned by the Kasetsart University, Bangkok. The mosquito fauna at this site include a number of dominant vector species, including *An. dirus* and *An. minimus* alongside their siblings *An. baimaii* and *An. harrisoni* respectively (Appendix C, Figure 11 and Table 6 show the exact species distribution). Mosquitoes were collected at night, carefully placed into large sample cups and recorded the following day using a high-spec Telinga EM23 field microphone and a budget smartphone (see Appendix D.3 for device details).

**Tanzania**    While Asia has the most diverse vectors, sub-Saharan Africa has the most dangerous mosquito species (*An. gambiae*), responsible for the highest transmission of human malaria in the world, and the highest number of deaths [World Health Organization, 2020]. In collaboration with the IHI, HBNs, larval collections and CDC-LTs (metadata `method`, Appendix C) were used to sample wild mosquitoes in the Kilombero Valley, Tanzania, and record them in sample cups in the laboratory. *An. gambiae* and *An. funestus* (another highly dangerous mosquito found across sub-Saharan Africa), are also siblings within their respective species complexes. Thus, standard polymerase chain reaction (PCR) identification techniques [Scott et al., 1993] were used to fully identify mosquitoes from these groups.[2] For all the cup recordings in Thailand and Tanzania, environmental conditions (temperature, humidity) were monitored throughout the recording process. The Tanzanian sampling has collected 17 different species (Figure 11, Table 6 show a full breakdown). Example spectrograms are shown for the eight most populated species in Appendix B.5 Figure 8.

## 5    Benchmark

To showcase the utility of the data, we supply baseline models for MED in Section 5.1, and MSC in Section 5.2. For both tasks, we discuss possible data biases arising from species imbalance, mosquito types, and multiple recording devices, and suggest mitigation strategies in Appendix D.6. Detailed instructions for code use are given in Appendix B. Further use cases are discussed in Appendix D.5.

---

[2]The database gives the PCR identification within the `species` column, or the genus/complex if not available.

**Models**  BNNs provide estimates of uncertainty, alongside strong supervised classification performance, which is desirable for real-world use cases such as ours. BNNs are also naturally suited to Bayesian decision theory, which benefits decision-making applications with different costs on error types (e.g. *Anopholes* species are more critical to classify correctly) [Vadera et al., 2021, Cobb et al., 2018]. We thus supply three benchmark BNN model classes for this dataset, noting that their equivalent deterministic counterparts achieved either equal or marginally worse classification performance. Details of the training hardware, hyperparameters, and modifications to the models are given in Appendix B.4.

1. **MozzBNNv2**: A CNN with four convolutional, two max-pooling, and one fully connected layer augmented with dropout layers (shown in Appendix B.4, Figure 3). Its structure is based on prior models that have been successful in assisting domain experts in curating parts of this dataset with uncertainty metrics [Kiskin et al., 2021].

2. **ResNet BNN**: ResNet has achieved state-of-the-art performance in audio tasks [Palanisamy et al., 2020] motivating its use as a baseline model in this paper. We augment the model with dropout layers in the building blocks to approximate a BNN. We opt to use the pre-trained model for a warm start to the weight approximations.

3. **VGGish BNN**: VGGish has become a benchmark in a variety of audio recognition tasks [Hershey et al., 2017]. We use the full pre-trained *features* and *embeddings* model, adding a single dropout and final linear layer to perform MC dropout for classification. We describe further modifications to the model class in Appendix B.4.

**Features**  We provide the following features for our models (see Appendix B.3 for details):

1. **Feat. A**: Features with default configuration from the VGGish GitHub intended for use with VGGish: 64 log-mel spectrogram coefficients using 96 feature frames of 10 ms duration forming a single example $\mathbf{X}_i \in \mathbb{R}^{64 \times 96}$ with a temporal window of 0.96 s.

2. **Feat. B**: Features originally designed for MozzBNNv2 (previous mosquito detection work [Kiskin et al., 2021]): 128 log-mel spectrogram coefficients with a reduced time window of 30 (from 40) feature frames and a stride of 5 frames for training. Each frame spans 64 ms, forming a single training example $\mathbf{X}_i \in \mathbb{R}^{128 \times 30}$ with a temporal window of 1.92 s.

**Performance metrics**  We define the test performance with four metrics: the receiver operating characteristic area-under-curve score (ROC AUC), the precision-recall area-under-curve score (PR AUC), the true positive rate (TPR), also known as the recall, and the true negative rate (TNR), to account for class imbalances in the test sets. These are evaluated over non-overlapping feature windows of 1.92 seconds. To compare the feature sets fairly, Feat. A test data is aggregated over neighbouring windows to form decisions over 1.92 s intervals. Edge cases where the data cannot be partitioned into full examples are removed from the test sets.

### 5.1  Task 1: Mosquito Event Detection (MED)

For mosquito event detection, we hold out Test A of labelled field data which most closely resembles the recording configuration of our system in Figure 1. Achieving good performance on that set does not guarantee good scalability to other use cases in itself. Therefore, we also evaluate over Test B, recorded in a cage placed in a highly noisy domestic environment. As a result, the SNR is much lower than that of Test A. The statistics of the training and test sets are given in the rows of Table 2.

For the intended use case of Test A, all of the model and feature combinations were able to achieve ROC AUC above 0.93 and PR AUC above 0.90 (Table 3). Furthermore, all of the models improve in performance when utilising Feat. A over Feat. B. However, performance on Test B is significantly lower for all models with no clear preference for features. The highest AUCs are achieved by BNN-ResNet when trained on Feat. B (ResNet18: ROC: 0.770, PR: 0.749, ResNet50: ROC: 0.76, PR: 0.750). To verify that the issue does not lie in the test set, after manually verifying each label resulting from feature extraction, we trained a model on half of Test B to achieve an ROC AUC of 0.915 on the second half of Test B. (Appendix B.5, Figure 4). Furthermore, prior work was able to achieve ROC AUCs of 0.871 to 0.952 with smaller neural networks which were optimised for use with scarce data [Kiskin et al., 2017]. The task presented in this paper, however, is to be able to achieve good performance over Test B, in addition to Test A, without the model having access

Table 3: **Mosquito Event Detection (MED)**. **Test A**: IHI Tanzania with HumBug Net. **Test B**: Oxford Zoology caged. Evaluated over $N_{\text{mozz}}$ mosquito, and $N_{\text{noise}}$ background 1.92 second samples. 30 samples drawn from each BNN to estimate the posterior. ROC AUC, PR AUC, TPR and TNR scores given as percentages ($\times 10^2$). The baseline ROC AUC score is given by 50 (completely random classifier). PR AUCs are relative to the prevalence of the classes, given by $N_{\text{mozz}}/(N_{\text{mozz}} + N_{\text{noise}})$.

| Data | Metric | MozzBNNv2 | | BNN-ResNet50 | | BNN-ResNet18 | | BNN-VGGish | |
| --- | --- | --- | --- | --- | --- | --- | --- | --- | --- |
| | | Feat. A | Feat. B | Feat. A | Feat. B | Feat. A | Feat. B | Feat. A | Feat. B |
| **Test A** | ROC | 98.1 | 96.4 | 98.3 | 93.0 | 98.1 | 92.5 | **98.5** | 97.3 |
| $N_{\text{mozz}}$: 1,714 | PR | 97.9 | 97.1 | **98.2** | 93.6 | 98.0 | 89.5 | 98.1 | 97.6 |
| $N_{\text{noise}}$: 2,068 | TPR | 79.5 | 79.9 | 76.9 | 79.1 | 67.0 | 76.1 | 85.6 | **87.3** |
| | TNR | 98.3 | 98.4 | **99.0** | 91.2 | 99.5 | 89.1 | 98.4 | 97.4 |
| **Test B** | ROC | 71.1 | 58.4 | 74.8 | 76.1 | 71.1 | **77.0** | 74.1 | 57.4 |
| $N_{\text{mozz}}$: 616 | PR | 64.0 | 63.2 | 72.0 | **75.0** | 68.5 | 74.9 | 70.7 | 61.3 |
| $N_{\text{noise}}$: 1,084 | TPR | 30.1 | 30.9 | 31.0 | **34.1** | 30.6 | 32.8 | 30.8 | 31.7 |
| | TNR | 99.3 | 99.2 | **100.0** | 98.8 | **100.0** | 99.3 | **100.0** | 99.3 |

to any data (or covariates) from either test set during training. This task therefore poses a challenge to promote the development of generalisable deep learning models, which we require for robust deployment.

## 5.2 Task 2: Mosquito Species Classification (MSC)

This task utilises data collected with a wide range of well-populated species of wild captured mosquitoes at IHI Tanzania. We split the 8 most populated species by recordings (each `audio_id` records a unique mosquito) into a 75-25 % train-test partition through a range of 5 fixed random seeds. To address data imbalance, upon training, we supply class weights as the inverse of the class frequency. From our experiments, this strategy has produced better results versus downsampling majority or oversampling minority classes, but there is likely room for improvement to be found here with paradigms such as few-shot learning [Sun et al., 2019], loss-calibrated inference [Cobb et al., 2018], and many more. To further motivate our two-stage pipeline, we note that the start and stop time tags for this dataset were auto-generated with a prior BCNN [Kiskin et al., 2021]. These factors contribute to a realistic test-bed for our pipeline of Figure 1, and hence any models developed for this dataset are candidates for real-world deployment.

The ROC AUC of 0.927 and PR AUC of 0.716 produced for this classification problem (Table 4) by the best-performing baseline model, MozzBNNv2-FeatB, demonstrate the ability to discriminate between different species of mosquitoes that have been sampled individually in the wild.

The results also show how our dataset is well suited for training multi-species classifiers to a degree that was not available previously. From the total ROC and PR AUCs, there is a slight preference for Feat. B for all models, except VGGish (as Feat. A were naturally made to be used with the model).

When interpreting PR AUC scores, a good indication of model performance is given by the increase in PR AUC over the baseline prevalence, given in the first column of Table 4. Due to the heavy class imbalance, the PR AUC scores are significantly lower on the minority classes, except for *Ae. aegypti* mosquitoes, which may be due to their larger size and hence more distinct difference in acoustic properties. The model confusion occurs in species with similar physical characteristics (see Appendix B.5, Figure 8 for a visualisation of spectra for each species). Example class-specific softmax outputs, ROC and PR curves, as well as confusion matrices are discussed in further detail in Appendix B.5.

Maximising PR performance of the under-represented, lower-scoring, classes, is the primary area in need of improvement in this task, which we encourage researchers to explore further.

Table 4: **Mosquito Species Classification (MSC):** Statistics, ROC AUC and PR AUC scores on the cup recordings conducted at IHI Tanzania. The total AUCs are given by the micro average. The baseline ROC AUC score is given by 50 (completely random classifier). PR AUC scores are relative to the prevalence of the classes, given by the number of (test) mosquitoes per class divided by the total number of mosquitoes (test). All scores are reported as mean (standard deviation) over 5 random train-test partitions ($\times 10^2$) of unique wild *'mosquitoes'*, with the distribution of column 1 in the form of train (test), prevalence (%).

| *Mosquito* Train (test), Prevalence | Metric | MozzBNNv2 | | BNN-ResNet50 | | BNN-ResNet18 | | BNN-VGGish | |
|---|---|---|---|---|---|---|---|---|---|
| | | Feat. A | Feat. B | Feat. A | Feat. B | Feat. A | Feat. B | Feat. A | Feat. B |
| *An. arabiensis* | ROC | 83.7 (1.2) | **86.6 (1.0)** | 75.8 (7.3) | 84.9 (2.4) | 75.6 (7.7) | 83.4 (8.7) | 85.7 (2.2) | 84.1 (1.5) |
| 385 (129), 36% | PR | 77.5 (2.5) | **80.9 (1.6)** | 71.8 (5.8) | 80.3 (4.4) | 67.9 (9.7) | 78.5 (8.8) | 80.2 (3.9) | 77.3 (2.2) |
| *Culex pipiens* | ROC | 81.4 (1.2) | **86.7 (1.4)** | 85.0 (2.2) | 84.0 (3.3) | 85.0 (2.5) | 85.6 (4.8) | 82.1 (1.7) | 81.4 (1.6) |
| 252 (84), 24% | PR | 57.3 (3.3) | 66.9 (2.3) | 61.4 (4.4) | 60.1 (5.6) | 60.3 (7.6) | 67.6 (8.3) | 59.0 (3.6) | 59.0 (3.0) |
| *Ae. aegypti* | ROC | 95.0 (0.8) | 96.4 (1.9) | **98.8 (0.6)** | 97.1 (1.8) | 98.2 (0.3) | 94.5 (1.1) | 96.6 (1.0) | 96.3 (2.3) |
| 36 (13), 3.6% | PR | 53.8 (7.2) | 74.4 (5.1) | **83.0 (2.7)** | 78.0 (11) | 76.6 (3.9) | 75.9 (3.1) | 66.6 (7.7) | 76.0 (4.9) |
| *An. funestus ss* | ROC | 91.7 (0.6) | 92.3 (1.3) | **93.8 (2.1)** | 84.7 (7.2) | 85.5 (7.7) | 90.6 (4.9) | 93.5 (1.4) | 91.0 (1.5) |
| 186 (62), 17.5% | PR | 78.2 (1.9) | 80.9 (1.1) | **84.6 (4.5)** | 70.9 (10) | 67.2 (14) | 77.4 (9.6) | 83.3 (3.3) | 76.0 (4.2) |
| *An. squamosus* | ROC | 78.2 (1.9) | 85.2 (2.4) | **88.8 (4.4)** | 85.2 (5.3) | 86.5 (3.2) | 83.5 (3.9) | 83.6 (3.3) | 86.4 (2.9) |
| 68 (23), 6.5% | PR | 21.1 (3.3) | 35.6 (5.8) | 39.4 (10) | 34.5 (8.5) | 36.0 (6.2) | **40.3 (9.8)** | 28.6 (8.1) | 35.6 (6.1) |
| *An. coustani* | ROC | 90.8 (2.3) | 88.4 (3.2) | **93.4 (1.4)** | 85.1 (4.6) | 92.2 (2.3) | 83.6 (5.5) | 89.9 (4.6) | 85.2 (4.1) |
| 37 (13), 3.6% | PR | 32.7 (8.0) | 26.6 (8.4) | **35.2 (8.5)** | 23.4 (11) | 32.5 (16) | 26.4 (9.8) | 33.2 (10) | 25.7 (8.2) |
| *Ma. uniformis* | ROC | 82.5 (7.6) | 82.0 (6.4) | **84.7 (6.9)** | 83.6 (9.4) | 87.5 (4.5) | 80.1 (8.8) | 83.4 (2.2) | 77.2 (8.3) |
| 57 (19), 5.4% | PR | 33.9 (8.7) | 29.6 (9.0) | 35.4 (10) | 34.5 (13) | **35.9 (7.8)** | 35.4 (13) | 29.1 (4.5) | 23.4 (5.2) |
| *Ma. africanus* | ROC | 91.2 (3.0) | 91.3 (1.7) | **93.0 (2.4)** | 84.5 (8.9) | 89.9 (4.6) | 85.8 (4.3) | 92.0 (2.6) | 91.1 (2.2) |
| 28 (10), 2.8% | PR | 26.8 (9.7) | 22.3 (5.0) | 29.0 (10) | 22.7 (19) | 24.3 (11) | 21.9 (4.2) | **33.5 (8.8)** | 23.4 (3.2) |
| | | | | | | | | | |
| **Total** | ROC | 91.4 (0.8) | **92.7 (0.9)** | 89.9 (2.5) | 90.4 (2.1) | 90.1 (2.1) | 90.8 (3.1) | 92.1 (1.2) | 91.4 (0.7) |
| 1049 (353) | PR | 66.9 (2.1) | **71.6 (2.2)** | 63.4 (4.8) | 65.0 (3.8) | 57.7 (7.3) | 69.2 (8.4) | 68.1 (3.9) | 66.2 (2.0) |

## 6 Conclusion

In this paper we present a database of 20 hours of finely labelled mosquito sounds and 15 hours of associated non-mosquito control data, constructed from carefully defined recording paradigms. Our recordings capture a diverse mixture of 36 species of mosquitoes from controlled conditions in laboratory cultures, as well as mosquitoes captured in the wild. The dataset is a result of a global co-ordination as part of the HumBug project. Our paper makes the significant contribution of providing both the large multi-species dataset and the infrastructure surrounding it, designed to make it straightforward for researchers to experiment with.

Despite decades of work, mosquito-borne diseases are still dangerous and prevalent, with malaria alone contributing to hundreds of thousands of death each year. Therefore a further contribution of this work is to make available mosquito data that is still a scarce commodity. In addition, we have highlighted that our dataset contains real field data collected from smartphones, as well as varying background environments and different experimental settings. As a result, this multi-species data set will continue to help domain-experts in the bio-sciences study the spread of mosquito-carrying diseases, as well as the myriad of factors that affect acoustic flight tone.

Finally, HumBugDB will be of interest to machine learning researchers working with acoustic data, both in the challenges posed by real-world acoustic data, as well as in the way that we use Bayesian neural networks for mosquito event detection and species classification. We provide baseline models alongside extensive documentation. As a result, we make it easy for researchers to start building their own models. It is our aim, by releasing this dataset and identifying areas for improvement in our baseline tasks, to encourage further work in the detection of mosquitoes. We hope this in turn leads to improved future detection and classification algorithms.

## Acknowledgments and Disclosure of Funding

This work has been funded from a 2014 Google Impact Challenge Award, and has received support from the Bill and Melinda Gates Foundation, [#opp1209888] since 2019. We would like to thank Paul I Howell and Dustin Miller (Centers for Disease Control and Prevention, Atlanta), Dr. Sheila Ogoma (The United States Army Medical Research Unit in Kenya (USAMRU-K)). Prof. Gay Gibson (Natural Resources Institute, University of Greenwich) and Dr. Vanessa Chen-Hussey and James

Pearce at the London School of Tropical Medicine and Hygiene. For significant help and use of their field site Prof. Theeraphap Chareonviriyaphap and members of his lab, specifically Dr. Rungarun Tisgratog and Jirod Nararak (Dept of Entomology, Kasesart University, Bangkok) and Dr. Michael J. Bangs (Public Health & Malaria Control International SOS Kuala Kencana, Papua, Indonesia). We also thank nVIDIA for the grant of a Titan Xp GPU.

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
