# OpenReview forum: "HumBugDB: A Large-scale Acoustic Mosquito Dataset"
_NeurIPS.cc/2021/Track/Datasets_and_Benchmarks/Round2 — NeurIPS 2021 Datasets and Benchmarks Track (Round 2)_

### Official Review · Reviewer_YrHc · 2021-09-19
**Review of HumBugDB**

**Rating:** 8
**Confidence:** 4
**Clarity:** The paper is well written and easy to…

**Strengths:**

The HumBugDB dataset is the largest and most diverse dataset currently available, with stronger labels than prior work.

The Authors provide a clear explanation of why mosquito detection and classification is an important task, and presents blueprints of a pipeline of how audio samples of mosquitos can be crowdsourced using smartphone. This is an important tool for more accurately predicting e.g. malaria or dengue outbreaks.

The authors clearly document the entire dataset construction process, the entire benchmarking process and gives insights into the results obtained.

The authors mention that voice activity detection and removal is used in future releases to ensure privacy. In the current release data has been curated to avoid voices unless explicitly allowed by the person in question.

The dataset is a clear contribution to the machine learning field as an interesting application for audio event detection and classification.


**Weaknesses:**

The paper solely evaluates using Bayesian Convolutional Neural Networks. This is due to the BNNs fitting well into decision making applications due to the provided uncertainty estimations. However, it would be interesting to see whether performance would improve if classic CNNs are used, at the cost of uncertainty estimation.

The paper evaluates the binary mosquito detection task and the multi-class mosquito species classification using the ROC-AUC. However, the data is stated to be imbalanced. Therefore, the ROC and ROC-AUC is not the appropriate metrics to use. Instead it would be more appropriate to use the Precision-Recall curves (see https://www.biostat.wisc.edu/~page/rocpr.pdf) or a metric such as Matthews Correlation Coefficient which considers the ratio of False/True Positive/Negatives.

It is unclear from the main manuscript whether there is a validation split. When examining the supplementary materials, it appears a random subset is used for validation for mosquito detection, while no subset is used for mosquito species classification. Could the authors explain why a static validation subset is not used, such that everyone uses the same validation data, or why cross-validation was not used


**Additional Feedback:**

it would be beneficial to include visual examples of your data. This could be spectrograms highlighting the differences between audio clips with and without mosquitos, or different kinds of mosquitos. This would fit into the supplementary materials.

**Correctness:**

The claims appear to be correct. The dataset construction process is clearly described and the dataset is benchmarked using three models and two feature descriptors. The benchmarking process is described in detail in the supplementary materials.

**Documentation:**

The authors provide a very detailed datasheet in the supplementary materials.
The dataset is hosted on zenodo with a version controlled DOI, where it is publicly available. Similarly, all code is hosted on a github repo. All licenses are also clearly stated.
In the manuscript a pipeline is proposed which would lead to a sustainable crowdsourcing effort.
The authors discuss ethical use and propose measures to remove human voice from submitted data.


**Ethics:**

Not that i am aware off.

**Relation To Prior Work:**

The paper clearly positions itself compared to prior work and explains how it is novel.

**Summary And Contributions:**

The paper presents an enormously interesting dataset on the problem of mosquito detection and species classification. The dataset is so far the largest public dataset with 20 hours of mosquito samples and 15 hours of background noise, all expertly labeled.

---

> ### Author Response · Authors · 2021-09-26
> **Thank you for your comprehensive review -- our actions to incorporate your suggestions**
>
> Thank you for your kind, well-structured and constructive review. We will address your comments and describe the changes we will implement based on your helpful feedback.
>
> ### Weaknesses
>
> > The paper solely evaluates using Bayesian Convolutional Neural Networks... However, it would be interesting to see whether performance would improve if classic CNNs are used, at the cost of uncertainty estimation.
>
> This is a very interesting point that we considered: the dropout components of the BNNs are simple to remove from the code (and the evaluation code functions by default for vanilla CNNs as documented in the supplementary material), and we have tested the models in early experiments. We achieved slightly worse performance with vanilla CNNs, though the main benefit of BNNs as you correctly identify is the uncertainty estimation. We achieve equivalent or better performance with BNNs, with the only drawbacks of slightly increased training time (due to more regularisation during training), and a linear cost increase in evaluation (the number of samples drawn from the network). **We will clarify this in the paper.**
>
> > The paper evaluates the binary mosquito detection task and the multi-class mosquito species classification using the ROC-AUC. However, the data is stated to be imbalanced. Therefore, the ROC and ROC-AUC is not the appropriate metrics to use.
>
> Thank you for pointing this out and helping us further improve our work. Our data for the MED task is fairly well balanced (ratios of ~17:20, ~3:5 for Test A and Test B respectively), and the ROC AUC accurately reflects model performance.
>
> For the MSC task, the strongest imbalance in the test set is a factor of ~35:1, and occurs for a few minority classes. We note that the PR curves are sensitive to the ratio of samples per class (rebalancing the test via undersampling affects the relative scoring of PR AUC curves more than ROC). **We are therefore working on adding PR AUC scores and baseline prevalences for both tasks for completeness (without removing ROC-AUC) to better describe the overall performance of the models.  We will also detail in the supplement (B5: Species Classification) a more nuanced discussion on PR/ROC performance, visualise example PR curves, and demonstrate the softmax outputs per class over the test set.**
>
> > It is unclear from the main manuscript whether there is a validation split. When examining the supplementary materials, it appears a random subset is used for validation for mosquito detection, while no subset is used for mosquito species classification. Could the authors explain why a static validation subset is not used, such that everyone uses the same validation data, or why cross-validation was not used.
>
> * MED: We used a fixed random seed for the validation subset, and it is included in our code as such, so that everyone uses the same validation by default. We note that we do not want to impose a fixed validation set for other researchers, as a better choice of validation data may result in improved performance over our baselines.
> * MSC: Due to the size of the datasets (especially compared to MED), we chose not to use validation data here, instead splitting the dataset by unique recordings across 5 (fixed) random seeds. This is a different strategy to withholding entire experimental groups for MED due to better data availability.
>
>
> > it would be beneficial to include visual examples of your data. This could be spectrograms highlighting the differences between audio clips with and without mosquitos, or different kinds of mosquitos. This would fit into the supplementary materials.
>
> Thank you for the suggestion. **We will include spectral representations of individual mosquito species from subsets of the database, and also an illustration of audio clips without mosquito for the supplementary materials.**
>
> We will endeavour to make the changes as soon as possible, at the latest before the camera ready version. Thank you once again for identifying very useful areas of improvement.
>
> Kindest regards,
>
> Paper 132 Authors.

---

### Official Review · Reviewer_xg74 · 2021-09-20
**Robust dataset of audio recordings of mosquitoes**

**Rating:** 8
**Confidence:** 3
**Clarity:** The paper is well written, and the co…

**Strengths:**

The main strength of the contribution lies in the large scale of the dataset, which includes 20 hours of mosquitoes recordings from different locations, with an average duration of audio samples of 9.7 seconds, which is significantly longer than samples from previous datasets proposed for the same purpose. The number of species is also larger than previous studies. In addition, the audio samples have been recorded in controlled conditions, that increase the reliability of data and labels.

The contribution is accompanied by a comprehensive datasheet that provides valuable information about the dataset, reinforcing the seriousness of the paper and the soundness of the dataset.

**Weaknesses:**

My main concern on the paper is about the technical challenges the dataset may bring to the machine learning community. It is not clear what are the limitations of current techniques, in particular those developed for acoustic event detection and sound classification, and in which manner these techniques would not be able to address the tasks discussed in the paper.

The baseline models provided by the authors seem to indicate that good performance are already possible, and it is not unreasonable to think that extended analysis of the data, using for example more systematic fine-tuning of hyperparameters, data augmentation strategies, expert knowledge, etc., may bring better results relatively easily.

Therefore, the contribution might be more suitable for application domain specialists (such as entomologists) that would have an interest in connecting results obtained using standard machine learning techniques with domain expertise to gain knowledge about the health issues related to mosquitoes.  Machine learning researchers, who constitute the main audience of the NeurIPS conference, might not find in this dataset the technical barriers that will lead to the development of more sophisticated models and techniques.

**Additional Feedback:**

The paper is a resubmission from Round 1 that has been significantly improved based on the comments of the reviewers of Round 1. In my opinion, the comments have been carefully addressed, and the dataset could be used by researchers to investigate this important problem. My main limitation concerns the relevance of the dataset for the machine learning community as discussed in the Weakness section, and I would be happy to hear from the authors about the technical barriers connected to the dataset they identified in terms of methodology, beyond the application domain.

Update after discussions with authors: I have reconsidered my position regarding the relevance of the dataset for the conference (see comments below).


Other comments
--------------
* The first sentence states that mosquitoes are not present in Antarctica, with a reference to a 2013 publication. Is there any more up-to-date publication to confirm this is still the case with the current rise of temperature?
* According to WHO website, numbers for malaria are '219 millions' and 400 000 deaths, in disagreement with the numbers reported in the introduction.
* Given a system that accurately detect and classify mosquitoes from a sound recording trained using the dataset, what would be the typical use of such a system in real-world conditions? In uncontrolled environment, it seems very unlikely that a mosquito will fly close enough to the microphone to trigger a detection. In that case, would you say the dataset would only be useful in the same conditions as the data collection, i.e., capturing mosquitoes and recording them?


**Correctness:**

The submission consists of a dataset and of two benchmarks. The dataset is constructed in a sound way, and has the potential to lead to the development of machine learning models that may help researchers interested by such application.

The benchmarks consist in two tasks related to the dataset. The level of details provided and the code accompanying the paper should be sufficient to reproduce the results, even if I haven't tried myself to do it.
The claims made by the authors seem correct and reasonable, considering the information provided in the submission.

**Documentation:**

There is sufficient detail on the dataset thanks to a comprehensive datasheet accompanying the paper. The dataset is publicly available, and can be downloaded in an easy way, and has a structure that makes its use relatively simple.

The code is available on a dedicated Github, with sufficient information to install the environment and run the code. Re-using the code might be a bit tedious because of the lack of document in the script files, and the presence of commented lines without explanation about their role, but stays largely possible.

**Ethics:**

Ethical considerations about the privacy of staff members doing the experiments is discussed, as samples may include voice recording and "other types of personal data". It would be interesting to know what the authors mean by "other types" to verify that the dataset is indeed respectful of privacy on this aspect. The authors claim that all persons whose voice is in the dataset have given their consent.

Regarding voice data, processing has been applied to remove samples with voice from persons who do not gave their consent. The use of a Google service to perform this task could be discussed to understand if this is compatible with the processing of personal data. No evaluation of the procedure is provided.

Beyond that, I do not see any additional ethical aspects regarding the dataset.

**Relation To Prior Work:**

The authors included a discussion of related works, and compared the characteristics of their dataset with previous ones.

**Summary And Contributions:**

This contribution presents a dataset that consists of 20 hours of audio recordings of mosquitoes from different species tracked in free flight. Audio recordings have been realised in different locations around the world, with various settings and environments.

In relation with the dataset, two tasks are suggested, acting like benchmarks to foster research on the topic:
* Identification of mosquitoes, i.e., binary classification of the presence or not of a mosquito in an audio recording.
* Classification of detected mosquitoes into one of the 36 different species.

The elaboration of the dataset is justified as mosquitoes are involved in the propagation of diseases. An automated way to detect and classify mosquitoes would have a significant impact on the management of the risks they pose, and increase the health conditions of affected regions.

---

> ### Author Response · Authors · 2021-09-26
> **Thank you for your review: clarifications and improvements from our end**
>
> Many thanks for your detailed review and appreciating the scale of this work. We are happy to provide answers to your questions and suggestions, and update some sections of the text to reflect these points more clearly.
>
> > It is not clear what are the limitations of current techniques, in particular those developed for acoustic event detection and sound classification, and in which manner these techniques would not be able to address the tasks discussed in the paper.
>
> **MED**: *“The highest ROC of 0.770 is achieved by BNN-ResNet18 when trained on Feat. B.”*
>
> This corresponds to a true positive rate of 0.328 only.
>
> **MSC**: We supply a typical confusion matrix (Appendix B5) corresponding to the best performing model, which upon inspection reveals that there is a lot of room for improvement. **We are adding Precision-Recall AUC scores to the tables to make it clearer the deficiencies of the models in class-specific performance.**
>
> **We would also like to add a limitations section after 5.2 to highlight the areas in need of improvement (in the fields of audio event detection), and add further detail in the supplement B5: Species Classification.**
>
> > Machine learning researchers, who constitute the main audience of the NeurIPS conference, might not find in this dataset the technical barriers that will lead to the development of more sophisticated models and techniques.
>
> We politely disagree, noting that there are very difficult technical barriers to overcome which require advancement in the state of art beyond data augmentation and careful hyperparameter tuning:
>
> The key point is that our data is constructed in a way where the test data is truly separate (out of sample and out of distribution) from the training data, presenting a realistic scenario for developing machine learning models that promote success for real-world deployment. This requires learning models that perform consistently across many data partitions:
>
> **MED**: Our baselines struggle on MED: Test B, despite access to plentiful training data. We have also demonstrated strong performance in the past when trained on a subset of that dataset, but are unable to trivially train a deep learning model which generalises to this dataset when entirely withheld during training.
>
> **MSC**: Classification is performed in an imbalanced data regime. There is potential for few-short learning, self-supervised learning, and other paradigms to be able to achieve better performance, in particular for the data-sparse classes. **This will become clearer when we add PR-AUC scores to the text**.
>
> ### Documentation
>
> We agree on this point, and will edit the script files to include better documentation within the main functions themselves. To add to this, we have extensively documented the code in Appendix B5, as well as created multiple Readmes in the repo to help with each task.
>
> ### Ethics
> By *"other types of personal data"* we meant that we withheld some of the data where children crying/singing were picked up by the bednet microphones during the study. As the study collected thousands of hours of data, there may be other unlabelled data with acoustic information from humans which we are not aware of (as we do not listen to all of the audio). **It is for this reason that we released only explicitly labelled sections**. Some of those include human speech recorded from entomologists (authors of this paper) with their consent.
>
> In the paper we state that *“To ensure no speech that has not had explicit consent for is included in future releases, we perform voice activity detection (VAD) and removal using Google’s WebRTC project...”*
>
> In the current release, no VAD was performed. We note we use WebRTC as a standalone GitHub repository with the code not having any access to internet servers, and it does not process any personal information or require any sort of subscription.
>
> **As for the evaluation, we will update our submission to include basic figures on the accuracy of the VAD method.**
>
> ### Other comments
>
> * We will add a reference from 2016 which categorically states “Continental Antarctica is devoid of insects" (https://www.annualreviews.org/doi/10.1146/annurev-ento-010715-023537). The rise of temperatures have some effect on insect population, but as of yet there are no suitable food sources or living conditions for mosquitoes.
>
> * WHO comment: Thank you for spotting this, our figures are correct but pointed to the wrong source, see the correct source here: https://www.who.int/news-room/fact-sheets/detail/malaria. **We will update the citation to the World Malaria Report of 2020.**
>
> * This paper has a mixture of sources: Test 1A specifically is recorded in a semi-field facility which mimics local housing construction: our target area of deployment. We describe our system of attracting mosquitoes in Section 3 (lines 134+), and the data captured with this system forms Test 1A (real-world conditions).
>
> We are very happy to respond to any further queries you may have.

---

> > ### Comment · Reviewer_xg74 · 2021-09-29
> > **Response to the clarifications made by the authors**
> >
> > Thank you very much for your responses to my review. I am glad to see that my comments have been carefully taken into account to further improve the quality of the publication and of the data and code accompanying it.
> > Regarding my main issue about the relevance of the dataset for a machine learning conference, the additional elements provided in the response have not completely changed my mind, but I do realise that, despite my negative assessment, there might be a positive impact, even limited, also on methodological developments, making this submission adapted to the conference.
> > I will therefore reconsider my position and update my review accordingly.

---

> > > ### Author Response · Authors · 2021-09-29
> > > **Thank you for your update**
> > >
> > > We are very grateful for your reply and updated review: we are in the process of implementing all of the changes we suggested based on all of the valuable reviewer feedback.
> > >
> > > Best wishes,
> > >
> > > Paper132 Authors.

---

### Official Review · Reviewer_VjrA · 2021-09-22
**A Large-scale Acoustic Mosquito Dataset**

**Rating:** 9
**Confidence:** 4
**Clarity:** The writing of the paper is clear, ma…

**Strengths:**

The dataset introduced in this paper seems to bring a significant contribution to the scientific community. Some strengths worth mentioning are the following:

1. The dataset was created in collaboration with mosquito entomologists, making it relevant not only from the machine learning point of view but also from the bio-science one.

2. The authors mention this is the first large-scale multi-species dataset of acoustic recordings of mosquitoes tracked continuously in free flight.

3. It contains 36 species of mosquitos captured from multiple locations on the Globe, such as the UK, USA, Thailand, or Tanzania. Furthermore, it includes both laboratory culture mosquitoes and wild captured ones, all these aspects providing variety.

4. It contains both noise-free sounds and real-world audio captured with low-cost smartphones and professional devices, allowing the study of the behavior in both scenarios.

5. Extensive experiments are conducted on the dataset using performant audio models for mosquito event detection and mosquito species classification, providing, therefore, a notable benchmark.

**Weaknesses:**

I did not notice any significant weaknesses.

One thing that would be nice to be added is a pie chart revealing the number of mosquitoes per species or a study of the class balance in the dataset. As the authors mention performing weighted classification instead of the classical one, an additional discussion on this matter would better position the dataset.

**Additional Feedback:**

None.

**Correctness:**

The dataset is constructed soundly by a team including both machine learning experts and entomology scientists. Therefore, the technical aspects are fully informed.

**Documentation:**

The data collection protocol is clearly described together with its applications. In addition, the dataset is publicly available, as well as the code and the pretrained models presented in the paper. Everything is well explained and documented.

**Ethics:**

The paper contains a privacy section, mentioning that full consent was obtained from the people whose voices may appear in the dataset. As no other personal information is included, I assume there is no need to perform an additional ethical review.

**Relation To Prior Work:**

The related work section is well documented, and the advantages of this dataset over the existing ones are clearly described.

**Summary And Contributions:**

This paper introduces a large-scale acoustic dataset containing around 20 hours of audio recordings of 36 different mosquito species. It also proposes two models based on Bayesian CNNs, one for mosquito event detection (MED), aiming to identify mosquitos from the overall background sound, and one for mosquito species classification (MSC). The dataset is mentioned to be the first large-scale multi-species one tracking mosquitos continuously in free flight.

---

> ### Author Response · Authors · 2021-09-26
> **Thank you for your highly positive review and suggestions**
>
> Thank you for your highly encouraging review. We are pleased that you have identified the scale and significance of this work. We will respond to your constructive suggestions to further solidify our work:
>
> > One thing that would be nice to be added is a pie chart revealing the number of mosquitoes per species or a study of the class balance in the dataset.
>
> Thank you for the suggestion: Figure 7 and Table 6 of the supplementary material show the species class balance per experiment group. We would be happy to turn Figure 7 into a pie chart instead. If space permits before camera ready acceptance, we would also be happy to move the visualisation into the main text to improve its visibility.
>
> > As the authors mention performing weighted classification instead of the classical one, an additional discussion on this matter would better position the dataset.
>
> We will expand on the section where we mention class weights (5.2) to describe:
> * Other options we have tried (class random over-sampling)
> * Extensions for more principled class weighting, especially with unequal costing (or utility) placed on classes
>
> Kindest regards,
>
> Paper 132 Authors.

---

### Decision · Program_Chairs · 2021-10-09

**Decision:**

Accept

**Comment:**

The data provided in the paper is of relevance for the NeurIPS data track. Based on reviewers opinions, and in particular after discussion with the authors, the paper achieves the minimum score required for publication at NeurIPS data track.